# Position: System-2 AI is about Complexity Out of Distribution

## Abstract

This position paper argues that addressing the limitations of the current "System-1" paradigm in deep learning, which struggles to generalize to complex scenarios beyond training, necessitates the introduction of a complementary "System-2" reasoning paradigm. We introduce the concept of "complexity out-of-distribution," which highlights the obstacles in progressing toward true artificial general intelligence (AGI). These scenarios require more intricate representations or computational paths than those encountered during training. Our position is that achieving effective solutions for such out-of-distribution complexities calls for a shift towards System-2, which frames problem-solving as a search over sequences of semantic units with unbounded complexity. This new paradigm seeks to discover algorithms, leveraging System-1's learned representations and heuristics, to handle examples with varying complexity akin to human reasoning abilities. We assert that advancements necessitate the development of tailored System-2 methods, including complexity-focused tasks, benchmarks, supervision paradigms, representations, metrics, and inductive biases. By drawing on recent research across multiple domains, we outline the essential requirements and challenges in integrating the symbolic search process of System-2 with neural network architectures.

## 1. Introduction

In recent years, the remarkable progress of artificial intelligence has largely been fueled by the paradigm of pre-training on massive datasets. This approach leverages abundant data and trains very large networks on them (Achiam et al., 2023; Radford et al., 2021; Ramesh et al., 2021).

[1]Anonymous Institution, Anonymous City, Anonymous Region, Anonymous Country. Correspondence to: Anonymous Author <anon.email@domain.com>.

Preliminary work. Under review by the International Conference on Machine Learning (ICML). Do not distribute.

However, just as fossil fuels are ultimately exhaustible, the availability of fresh, high-quality data is not infinite; we may soon face a point where there is no new data left for further pre-training (Villalobos et al.; Shumailov et al., 2024). Meanwhile, in many complex tasks, there remains a significant performance gap between AI systems and humans—one that cannot be closed simply by providing more data. The most prominent challenges in this regard are commonly referred to as reasoning or "System-2" tasks. Unlike "System-1," which relies on rapid, intuitive processing, System-2 problems require multi-step solutions where each step may either represent part of the final answer or serve as a pathway toward it (Kahneman, 2011). In reasoning tasks, the need for data grows exponentially, and to improve performance, we are compelled to scale data exponentially (Zeng et al., 2024). This intensifies the challenge, highlighting the limitations of relying solely on more data for AI advancement

A major objective of AI is to achieve correct prediction and generalization on unseen samples. Naturally, without certain assumptions—often called inductive biases—it is impossible to guarantee correct predictions on out-of-distribution (OoD) data (Mitchell, 1980; Baxter, 2000; Goyal & Bengio, 2022). Indeed, throughout the history of AI, every encounter with novel or unexpected data distributions has spurred researchers to identify and leverage new inductive biases. For instance, convolutional architectures with pooling layers embody the assumption of translational invariance, enabling more robust performance on spatially transformed inputs. However, this raises a key question regarding System-2 tasks: what kind of OoD challenge do such tasks pose?

The main characteristic of System-2 problems is that they require multi-step solutions that involve the composition of various subcomponents. **We posit that the primary out-of-distribution (OoD) issue related to System-2 problems is complexity out-of-distribution. Achieving effective solutions for such complexities requires a shift, which approaches problem-solving as a search over sequences of semantic units with unbounded complexity**. Complexity OoD refers to generalizing from a set of examples to new instances that require more complex representations or deeper computational paths. Crucially, no matter how large a given training set might be, there are always examples with higher complexity outside of it; as a result, even if

we had an infinite supply of data, simply scaling data size would not ensure the ability to handle increased complexity. This highlights a fundamental limitation of the "more data" strategy: complexity can escalate in ways that pure data scaling cannot accommodate

In this paper, we propose that System-2 problems should be approached from the perspective of complexity out-of-distribution and systematically analyzed under this framework. An approach that allows all System-2 problems to be examined within a unified framework.

In the following sections, we first define complexity. Then, we examine the format and structure of solutions for System-2 problems, explaining the concepts of atomic concepts and programs. After elucidating these two concepts, we discuss the connection and bridge between System-1 and System-2, and how they can aid each other. Finally, we address the research challenges in the area of System-2 and offer suggestions for constructing tasks, benchmarks, and establishing new supervisory paradigms.

## 2. Complexity Out of Distribution

As stated in the introduction, the primary characteristic of System-2 problems is that they require multi-step solutions. This multi-step nature of solutions is closely related to the concept of complexity. Here, we intuitively define complexity as the minimum number of sub-steps required to reach the solution. A complexity OoD setting then arises when test samples have a higher minimum number of sub-steps than any of the training examples, forcing the model to handle levels of complexity unaddressed by its prior experience. Examples that are less frequent or more complex are typically represented or processed by synthesizing multiple such atoms. This means that the more common cases are essentially learned through a System-1 approach, while the rarer or more complex cases require a System-2 solution that involves composing multiple learned building blocks, where each of these sub-steps is resolved as a System-1 task.

**Examples of Complexity OoD**: Consider the learning process for Roman numerals, where numerals like I, V, and X are frequently encountered and readily learned as discrete elements. These numerals are simple enough for the model to be processed straightforwardly through System-1, where the recognition is fast and reflexive. However, recognizing complex combinations or higher values like XXIV (24) or LXXXIX (89) or even IX (9) demands a greater cognitive integration of these simpler units, a task that calls upon the functionalities of System-2 to systematically combine these learned elements into a coherent understanding that respects the rules of Roman numeral composition. Similarly, for mathematical operations like multiplication, the model may learn basic single-digit or small multi-digit multiplica-

tion as "atoms" through System-1. But multiplying much larger numbers would necessitate combining these learned building blocks in a productive, systematic manner, which requires System-2 reasoning.

**Complexity OoD vs. Compositional OoD**: In the literature, two cases of compositional generalization are typically referred to as **systematicity** and **productivity**, respectively (Hupkes et al., 2020). The performance of models can be evaluated on these two out-of-distribution scenarios (Lake & Baroni, 2018; Hupkes et al., 2020; Loula et al., 2018). Systematicity refers to the ability to generalize to new combinations of known components during testing, even though those specific combinations were not present in the training data (Hupkes et al., 2020). Productivity refers to the ability to generalize to samples with greater length than those seen during training (Hupkes et al., 2020). While complexity OoD conceptually resembles compositional OoD, it differs from compositional OoD and its subcategories, systematicity and productivity. The distinction between the complexity OoD perspective and systematicity lies in the degree of complexity within the compositions. In systematicity, we seek generalization over combinations with a bounded number of components. In contrast, complexity OoD does not impose any bound on the number of foundational factors. The difference between the complexity perspective and productivity generalization lies in the focus area. In productivity generalization, the length of the input or output sequence is usually considered, whereas complexity OoD refers to a broader concept. For instance, one input might be shorter in length than another yet more complex, such as comparing the square root of 3 to that of 16.

**Representational and Computational OoD**: Complexity can be specified in two dimensions, representational and computational. Representational complexity OoD refers to samples that have more complexity than training samples. In fact, these are examples that, in order to be reconstructed or discriminated, require more detailed descriptions than the training samples. Another dimension of complexity that can be examined alongside representation is computational complexity. It pertains to samples for which the model needs additional processing steps to arrive at the correct answer, compared to the training samples. Upon reflecting, it becomes evident that the computational and representational dimensions are intertwined. Problems categorized under representational complexity OoD and computational complexity OoD are essentially two sides of the same coin, each influencing the other. The challenge, therefore, is not just to address these dimensions separately but to recognize that any successful solution must effectively integrate both aspects—representation and computation. It seems that to achieve System-2 and handle the variable amount of complexity in input samples, we need to solve both the problem of unlimited-length representation and have a mechanism

for adaptable-length computation. Here, we present a framework and worldview for a potential solution to System-2.

### 2.1. Program Synthesis Framework

We want a framework that is unbounded in its capacity to represent increasingly intricate input data and perform progressively more elaborate computations if required. Drawing from the ideas in the previous section, the more common or frequent samples can be learned as individual "atoms" through a System-1 approach during training. However, the rarer or more complex cases require a System-2 solution that involves systematically combining and reasoning with these learned building blocks in a productive and generalizable manner. Here, we aim to define what the solution format that System-2 is seeking actually is. Let's assume that we have a set of foundational semantic atomic units: $m_1, m_2, ..., m_n$. These foundational semantic units can serve as representation units (similar to words) in representation, or they can be basic functions (similar to basic mathematical operators) in computation. In that case, the solution that System-2 is seeking is finding the correct sequence of these foundational units. In the world of representations, this sequence can be likened to a sentence composed of words, while in the world of computations, it can be interpreted as an equation involving basic operators. If we also have an oracle that, given any sequence of units, performs the goal test and indicates whether the sequence is correct or not, then we can view the problem of finding the correct sequence as equivalent to a search problem. Let's introduce a familiar term for this setting: we define any possible sequence of units as a **program**. Programs can be correct or incorrect, and we are looking to search for the correct programs that are as short as possible.

### 2.2. Formal Definition of Complexity OoD

Below is a concise formalization of representational and computational complexity in terms of Kolmogorov Complexity (Kolmogorov, 1965; Li et al., 2008). Note that in practice, Kolmogorov Complexity itself is uncomputable; however, it serves as a useful theoretical lens for understanding what "complexity" might entail in System-2 reasoning.

#### 2.2.1. REPRESENTATIONAL COMPLEXITY OoD

Let $x$ be an input sample (e.g., an image, a sentence, or any structured data). We define its representational complexity using the notion of Kolmogorov Complexity, denoted by $K(x)$, which is the length of the shortest program (in a fixed universal programming language) that outputs $x$. Formally:

$$K(x) = \min\{|p| : U(p) = x\}, \qquad (1)$$

where $U(p)$ is the output of running program $p$ on a univer-

sal Turing machine $U$, and $|p|$ is the length (e.g., in bits) of that program. Intuitively, $K(x)$ measures how "complicated" or "rich" the description of $x$ is. When $K(x)$ is large, the sample $x$ has a high degree of structural or informational content, requiring more elaborate representation to describe it.

In a representational Complexity OoD scenario, a model is confronted during testing with samples $x_{\text{test}}$ such that

$$K(x_{\text{test}}) > \max_{x_{\text{train}} \in D_{\text{train}}} K(x_{\text{train}}), \qquad (2)$$

where $D_{\text{train}}$ is the set of training samples, and ">" informally indicates "significantly greater than." Thus, the representational complexity of $x_{\text{test}}$ exceeds any sample in the training distribution, requiring more detailed or intricate descriptions than the model has handled before.

#### 2.2.2. COMPUTATIONAL COMPLEXITY OoD

Let $y$ be the "output" or "solution" corresponding to input $x$. In System-2 tasks, $x \rightarrow y$ typically requires a multi-step reasoning or computational procedure. We capture the complexity of this procedure by the conditional Kolmogorov Complexity $K(y \mid x)$:

$$K(y \mid x) = \min\{|q| : U(x, q) = y\}, \qquad (3)$$

where now we consider programs $q$ that take $x$ as an input (or have $x$ hard-coded) and produce $y$ as output. Here, $|q|$ is the length of the shortest such program. If $K(y \mid x)$ is large, it signifies that the process of deriving $y$ from $x$ is inherently complex, involving many logical or algorithmic steps.

In a computational Complexity OoD scenario, a test pair $(x_{\text{test}}, y_{\text{test}})$ demands

$$K(y_{\text{test}} \mid x_{\text{test}}) > \max_{(x_{\text{train}}, y_{\text{train}}) \in D_{\text{train}}} K(y_{\text{train}} \mid x_{\text{train}}), \qquad (4)$$

implying that the minimal program needed to compute the solution from $x_{\text{test}}$ is transcending (in terms of length or algorithmic richness) the complexity observed for any training example. This goes beyond simply having more elaborate representations; it explicitly requires more computational steps or more complex logic.

## 3. System-1 Facilitates Achieving Solutions through System-2

From our definition, solving a problem with System-2 involves two key steps: (1) identifying the basic building

blocks (atomic units) and (2) combining them to form a valid solution. System-1 can offer valuable guidance for both steps:

1. How can we identify suitable atomic units?

2. How can we learn a heuristic function $h(p)$ or a program generator module to help System-2 find the correct program?

In both cases, learning is central. Because System-1 can learn to approximate virtually anything from experience, System-2 can use System-1 to approximate both the building blocks and the ways to combine them. This learning process is iterative: as System-1 improves, System-2's ability to discover and combine units improves as well, and vice versa.

### 3.1. Learning Proper Atomic Units

As discussed, a System-2 solution can be represented as a sequence (of non-fixed length) of semantic units. Each unit may be adjusted, and these units must together meet two conditions: 1) *Sufficiency*: any System-2 problem can be solved by combining these units; 2) *Minimal Redundancy*: no unit can itself be expressed as a program made of other units.

The agent begins by tackling simple tasks—attempting solutions with one-unit programs. A single unit (i.e., one-unit program) can be optimized through backpropagation to accomplish the task. This stage provides an initial approximation of the basic units.

Next, the agent attempts two-unit programs. At this point, either through searching over all two-unit combinations or via a program generator module, the agent identifies promising pairs and fine-tunes them jointly. This exposes the units to the ways in which they might collaborate. Notably, even if a unit was not useful by itself (in a one-unit program), it can become valuable when paired with others. This mutual refinement continues in the presence of more complex tasks and longer program lengths, ensuring that atomic units evolve in tandem with how they are best combined.

### 3.2. Learning a Heuristic Function to Generate Programs

Once we have established a suitable set of atomic units, System-2 still needs to synthesize a program from these units. One viable approach is to perform a search over possible programs. In the most extreme case, this search can be exhaustive, which will always find a correct solution (akin to "systematic generalization"). However, the deeper the correct program lies in the search tree, the longer this process may take.

Because time or computational resources are seldom unlimited, in practice both artificial agents and humans rely on System-1 "shortcuts" for search. For example, consider a scenario where a student is asked to prove a theorem in an exam. In theory, if given infinite time, this student could attempt every possible series of proof strategies—equivalent to an exhaustive search—and eventually arrive at a valid proof. However, during a timed exam, the student must rely on the heuristics and intuitions that reside in System-1 to efficiently narrow down potential approaches and arrive at a solution more quickly. This reliance on heuristics speeds up problem-solving but can also introduce errors or overfitting, representing a trade-off between guaranteed thoroughness and practical efficiency.

In human cognition, individuals with more sophisticated "System-1" intuitions can more swiftly generate accurate, innovative ideas, particularly in research or other creative domains. Over time, experts develop stronger heuristics in familiar problem spaces, reusing techniques in different combinations. Analogously, an agent's System-2 reasoning ultimately leverages such heuristic support from its learned System-1 modules. Thus, System-2's ability to combine atomic units effectively depends in large part on how well System-1 can (1) represent the relevant tasks and (2) generate promising configurations (or heuristics) to reduce the cost of search.

## 4. Some Shines of System-2 in Recent Researches

In recent years, there have been works that had the flavor or solution aligned with System-2 (as per our description). Numerous research studies have been conducted on various subproblems of System-2. These research efforts are akin to elephants in a dark room; in fact, each of them has touched upon a corner of the open issues for solving System-2 and attempted to address it, but they have not provided a comprehensive picture of the System-2 elephant. Here, we discuss those works and their relationship with our defined System-2, along with the necessary modifications for further advancement. We intend to translate several prominent works that have addressed each of the issues raised in System-2 here, under the same lens as our view of System-2.

### 4.1. Variable-length representation.

**Object-Centric Representation Learning:** While neural networks and common computer vision architectures perform reasonably well on regular images and tasks like image classification, they face challenges when dealing with images of complex scenes (Brady et al., 2023). In fact, as the number of details and objects in the scene increases, the representations provided by these networks become more flawed. In such complex scenarios, the phenomenon known

as the "superposition catastrophe" can occur (Von Der Malsburg, 1986; Greff et al., 2020). The superposition catastrophe refers to the issue where the representations learned by these networks fail to disentangle and separately encode the entities present in a complex scene. Superposition occurs in network representations because they have a fixed and limited length. The network must represent its understanding of the input image, whether simple or complex, within this fixed capacity. In recent years, researchers have increasingly focused on object-centric and structured representation learning, with "Slot Attention" being a notable method (Locatello et al., 2020). Slot Attention introduces an architecture that interacts with perceptual representations, like those from convolutional neural networks, to create abstract, adaptable slots. These slots can bind to any object in the input scene through a competitive attention mechanism. A key feature is that the number of output slots can be variable during inference, allowing for more slots as image complexity increases, thereby preventing superposition. Despite recent advances in object-centric representation learning, this field is still evolving and is struggling to address the challenge of obtaining more causal and compositional representations with less supervision (Didolkar et al., 2024; Mansouri et al.).

**Emergent Languages:** Language is one of the extraordinary skills of humans, through which they can communicate with other humans, have internal thoughts, and engage in reasoning. One of the branches of research in artificial intelligence is the field of Emergent Language (Havrylov & Titov, 2017; Lazaridou et al., 2022; 2018). In this branch, efforts are made to create conditions through games among multiple agents and enabling the exchange of messages between them, so that agents autonomously develop a constructed language. And then, after the emergence of the language, examine the compositional and linguistic characteristics of this newly created language (Lowe et al., 2019; Chaabouni et al., 2020). Some characteristics of the emerged language include its discrete words and also the non-fixed length of the transmitted messages. In fact, an agent for describing an input could generate a message with variable length (Ueda & Washio, 2021). This is related to one of the System-2 capabilities of humans. We can describe an image, no matter how complex it is, with more words as necessary. The discrete and variable-length nature of these messages as a representation closely corresponds to the view of program generation from learnable basic semantic units.

## 4.2. Variable-length computation

**Adaptive Computation Time:** One of the fundamental differences between humans and machine learning models is that the human response time to a problem can be a function of the difficulty of that problem, whereas, in machine learning models, the response time solely depends on the model architecture or the size of the input. For example, the longer the input sequence to a recurrent neural network (RNN), the longer it takes for the network to produce the final output. In other words, the human mind can devote more focus and attention to solving a problem with a more challenging input, something that traditional machine learning models are not capable of. To tackle this issue, Adaptive Computation Time (ACT), a mechanism embedding a halting unit within the RNN architecture, was introduced (Graves, 2016). This unit dynamically decides the number of computational steps for each time step by outputting a halting probability, allowing the RNN to either continue processing or move to the next step. This enhancement led to improved performance in tasks like binary vector parity, integer addition, and real number sorting. The concept of a halting mechanism was extended to the transformer architecture, resulting in the Universal Transformer, which improved performance and accuracy on various algorithmic and language understanding tasks (Dehghani et al., 2019).

**Learning to Program:** Symbolic regression is a problem in machine learning that aims to discover the underlying mathematical expressions or symbolic equations that describe a given dataset. Unlike traditional regression methods that rely on predefined functional forms (based on neural network architecture), symbolic regression attempts to find the symbolic expressions directly from the data. Symbolic regression has a close relationship with variable-length computation. This relationship arises from the fact that the mathematical expressions discovered by symbolic regression can have varying lengths and complexities, depending on the nature of the underlying relationship in the data (Biggio et al., 2021; Kamienny et al., 2022). This core idea was later more prominently implemented in the DreamCoder paper (Ellis et al., 2021) . Notably, in DreamCoder, subprograms that frequently co-occurred could be combined and refactored, simplifying the search process across different programs. Recently, during the 2024 Arc Challenge, a significant number of top-ranked solutions used the Program Generation approach (Chollet et al., 2024; Li et al., 2024b; Bonnet & Macfarlane, 2024; Ouellette, 2024).

## 4.3. Very recent advances based on Large Language Models (LLMs)

Another prominent place where the idea of program generation (mentioned in the above subsection) can be recently seen is in LLMs. Interestingly, an LLM can be seen as a program generator model that generates a step-by-step solution when given a problem as input. This program consists of tokens generated by the language model, which are, in a sense, executed by the model itself.

**Chain of Thought (CoT):** For reasoning tasks, LLMs can be asked to write the solution step-by-step before providing

the final answer (Wei et al., 2022) . This can enable the language model to generate longer solutions for more complex problems by generating tokens sequentially. The CoT idea helped significantly improve the performance of language models on some reasoning tasks. However, since LLMs are still confined to left-to-right decision-making processes (without backtracking) during inference, they can fall short in System-2 tasks that require exploration, strategic lookahead, or where initial decisions play a pivotal role. This means that for certain reasoning tasks, LLMs still faced challenges.

**LLMs and Search:** Since the trained LLMs by a System-1 approach can not guarantee to solve all reasoning problems naively by the CoT approach as discussed above, some approaches that need to explore during the test time in order to find the output have been introduced. Ideas such as Tree of Thought (ToT) and Graph of Thought (GoT) allow LLM to branch and generate the solution step-by-step through a search process during the inference time (Yao et al., 2024; Besta et al., 2023). ToT allows LLMs to perform deliberate decision-making by considering multiple different reasoning paths and self-evaluating choices to decide the next course of action, as well as looking ahead or backtracking when necessary to make global choices. In case of failure, it has the ability to backtrack and construct a new program. This concept is clearly analogous to the concept of learn-to-search.

**LLMs and repeated sampling:** LLMs, as probabilistic generative models, offer the capability to generate a diverse range of step-by-step solutions (or programs). LLMs achieve this through *repeated sampling*, a technique that increases the likelihood of generating an optimal response (Li et al., 2022; Rozière et al., 2023). Common sampling strategies in LLM inference include top-p (Nucleus Sampling) and top-k sampling, which enable the parallel generation of multiple candidate outputs. By leveraging repeated sampling, LLMs enhance their chances of producing accurate and high-quality responses, akin to how algorithm designers iteratively refine their solutions to improve computational efficiency. Self-correction is a test-time computation method that allows LLMs to iteratively revise and refine generated results using external or internal feedback (Shinn et al., 2023; Ye & Ng, 2024). A critical aspect of this iterative process is the implementation of evaluation and verification strategies, which ensure the effectiveness of repeated sampling and contribute to the overall reliability of the generated outputs. Selecting the most frequent answer as a verification strategy can enhance accuracy, particularly in approaches like self-consistency Chain of Thought (CoT), which improves mathematical reasoning accuracy by 18% compared to vanilla CoT (Wang et al., 2022; Li et al., 2024a; Lin et al., 2023) .

**LLMs and Reward Models:** Reward models are primarily categorized into two types: Outcome-based Reward Models (ORMs) and Process-based Reward Models (PRMs). ORMs utilize the accuracy of the final Chain of Thought output as feedback for training or verification (Cobbe et al., 2021). In contrast, PRMs are trained using feedback obtained from each individual reasoning step (Uesato et al., 2022; Lightman et al.). Some studies like (Lightman et al.) indicate that PRMs significantly outperform ORMs due to their more precise feedback which localizes any errors that occur. It is evident that PRMs require high-quality data and human feedback for each reasoning step during training (Lightman et al.). To tackle this problem, (Wang et al., 2024) proposes an automatic process annotation framework based on Monte Carlo Tree Search (MCTS). Reward models can essentially act as heuristic function to aid in the search among various step-by-step solutions which be viewed similar to the neural guided searches in program generation modules that have been introduced in Section 3.2. These models can be employed to discriminate between desirable and undesirable outputs when LLMs along with repeated sampling are used (to enhance search via rejection sampling). On the other hand, reward models can be employed in a Reinforcement Learning (RL) pipeline too.

**LLMs and RL**: A recent approach proposed in several studies (Wang et al., 2024; Setlur et al., 2024) is fine-tuning LLMs by an RL approach using the CoTs generated by the LLMs themselves and evaluated by verifiers or reward models (mentioned above) which provide supervision feedback. Unlike Supervised Fine Tunning (SFT) which tends to overfit to training data and struggles with generalization to out-of-distribution scenarios(Chu et al., 2025), RL generalizes to unseen situations more effectively. Despite the current community's preference for PRM-based verifier methods, especially following the success of the o1 model, a novel intuitive approach has recently been introduced in the DeepSeek R1 model (Guo et al., 2025). This approach eliminates the concept of verifiers entirely. Instead, during training, the model learns reasoning and step-by-step thinking through pure reinforcement learning (Guo et al., 2025; Wang et al., 2024). In fact, during the training phase, this model is trained using only two types of rewards: the correctness of the final answer and another that incentivizes structuring the reasoning process in a specific format. Remarkably, it achieves performance that is competitive with, and comparable to, the o1 model. This methodology demonstrates how RL can effectively facilitate program synthesis, significantly reducing or even eliminating the reliance on extensive search processes.

# 5. Constructing the Foundations of System-2 Learning Research

It should be noted that the explanations provided so far were merely in the context of formulating a new framework for System-2 and specifying its relationship with System-1. However, they are not sufficient for taking practical steps in the realm of System-2. To practically advance research in System-2, we need to establish practical foundations for it. These practical foundations are similar to everything that System-1 has traversed over the years: from fundamental elements such as tasks and datasets to benchmarks, as well as paradigms and more specific issues that build upon these foundations. Next, we aim to provide a general overview of what needs to be done for practical research in System-2. It should be reiterated that since we view System-2 in the context of the other side of the complexity out-of-distribution generalization coin, the general framework we observe is heavily based on this particular type of generalization and its associated challenges.

## 5.1. Tasks and Benchmarks

System-1 neural network architectures have existed for years, but the rapid evolution of deep learning began with the introduction of the ImageNet dataset in 2012. The event referred to as the ImageNet moment made the ImageNet dataset gain significant attention as the first large-scale dataset for benchmarking deep-learning vision networks. We believe that to ignite the progress of System-2, there must be a spark in creating tasks and benchmarks specifically tailored for it. In other words, System-2 needs its own version of the ImageNet moment. One example of such a benchmark is the ARC (Abstraction and Reasoning Corpus) Challenge proposed by François Chollet, which consists of tasks designed to evaluate more advanced reasoning capabilities beyond pattern recognition (Chollet, 2019). Of course, defining a foundational task with maximum inclusivity for System-2 is a non-trivial and complex matter, requiring extensive investigation. Nevertheless, alongside this main path, a parallel path can be pursued where tasks and benchmarks of System-1 are addressed using an approach inspired by System-2. For example, consider image classifiers that, upon receiving an image, attempt to generate the output over a variable number of steps depending on the complexity of the image.

## 5.2. Supervision Paradigms

Just like System-1, which includes supervised, unsupervised, and self-supervised learning, System-2 can also be categorized similarly. In System-1, supervision is based on the relationship between labels and features. In System-2, supervision involves both labels and programs. Essentially, while System-1 has a single network generating labels from fea-

tures, System-2 includes an additional program generation network. This network can be supervised or unsupervised regarding the presence of a program. It's natural that having supervision on programs helps us train the program generation module more efficiently (For example, consider a scenario where instead of showing examples to children for them to learn an algorithm, we directly teach them the algorithm or program itself). However, it's clear that not having supervision on programs is a scenario closer to reality. Therefore, we need a method for learning unsupervised programs as well.

Another promising direction for learning programs is to take a multitask learning and meta-learning approach. By exposing the system to a diverse set of tasks, it can potentially learn to discover the atomic units and compositional rules required for constructing programs. This process of learning to learn programs can be facilitated by meta-learning techniques that aim to acquire general program induction capabilities across a wide range of tasks. Such an approach aligns with the way humans learn to program by being exposed to various problem-solving scenarios, gradually developing an understanding of the fundamental building blocks and how to compose them to solve new tasks.

From here, we can follow a similar approach to System-1 for the program generation part (which is itself a System-1 module) and explore concepts like self-supervised learning for programs. Just like in System-1, we had different learning methods like supervised, unsupervised, and self-supervised learning, we can apply those same concepts to the program generation module in System-2. For example, we can explore self-supervised learning techniques specifically for learning programs without full supervision. This would be similar to how self-supervised learning was used in System-1 to learn representations from unlabeled data. The key point is that the program generation component itself can be treated as a System-1 module, and we can leverage the paradigms and methods developed for System-1 to improve how this component learns to generate programs.

## 5.3. Basic concepts need redefinition, such as representations and metrics

When defining System-2 by out-of-distribution complexity and foundational units for representation and computation, two challenges arise due to variable program lengths: suitable program representations and similarity metrics. Programs can vary greatly in length, making it crucial to design representations that capture their essence while accommodating this variability. After establishing appropriate program representations, the next challenge is defining a metric for program similarity. Traditional metrics like cosine similarity may not work due to programs' unique structures. A specific metric learning approach, such as using a Siamese

network or triplet loss, can train a model to minimize distances between similar programs and maximize those between dissimilar ones. Incorporating domain-specific knowledge—like structural similarity, function compositionality, or common substructures—can further refine program similarity assessments.

### 5.4. Other concepts that come into play again ...

When we built System-2 upon the foundation of System-1, the issues and challenges that System-1 faced are gradually reappearing with the development of System-2, albeit with a different appearance this time. For example, consider the problem of spurious correlation and shortcut learning (Arjovsky et al., 2019; Geirhos et al., 2020). In System-1, the situation was such that we encountered a problem in classification with input patterns having spurious correlations with labels. However, in System-2, we may face spurious correlation at two other reasoning levels: learning atomic elements and learning program generation. For example, consider a student facing a myriad of problems where addition is initially used to solve them. Now, the student may think that for every problem, the first step is to perform an addition operation. If the student encounters a problem that does not require addition, they may fall into the trap of spurious correlation and mistakenly perform addition.

### 5.5. And the biggest question: How can we include System-2 (include search) in neural networks?

A key challenge is integrating System-2 architectures with neural networks. Since the human brain integrates symbolic processing and search mechanisms with neural networks, these processes must also be grounded in neural networks for effective emulation. The reference to the brain being wired with neural networks implies that, in the human brain, both symbolic and intuitive processing are seamlessly integrated. If we aim to emulate similar capabilities in artificial systems, we must find ways to ground symbolic operations onto neural network structures. To address this issue, further studies in the field of neurocognitive science may be required.

## 6. Alternative Views

While the paper presents a compelling argument for integrating System-1 and System-2 paradigms to address complexity out-of-distribution (OoD) challenges, an alternative perspective is to consider learning and reasoning as largely independent processes, or at least as processes with minimal interaction. This view posits that the challenges associated with System-2 tasks should not necessarily be framed within the context of OoD complexity, nor should they be seen as an extension of System-1 capabilities.

1. **Independence of Learning and Reasoning**: In this view, learning (System-1) and reasoning (System-2) are distinct cognitive processes that operate independently. Learning is primarily about pattern recognition and the assimilation of information from data, while reasoning involves logical deduction and problem-solving that may not directly rely on learned patterns. This separation suggests that the development of reasoning capabilities does not need to be constrained by the limitations or structures of learning systems. Instead, reasoning can be approached as a standalone process, potentially leveraging symbolic logic, rule-based systems, or other non-neural methodologies that do not depend on the data-driven paradigms of System-1.

2. **Distinctness of System 2 and Complexity OoD**: The paper frames System-2 challenges as issues of complexity OoD, implying that reasoning tasks are fundamentally about handling more complex versions of learned tasks. However, reasoning can be seen as a fundamentally different type of cognitive activity that does not necessarily align with the OoD framework. Reasoning might involve abstract thinking, hypothetical scenarios, and counterfactual reasoning that do not fit neatly into the complexity OoD paradigm. These tasks may require entirely different approaches, such as symbolic reasoning engines or logic-based AI, which do not rely on the same principles as those used to address

## 7. Conclusion

Achieving artificial general intelligence (AGI) requires systems that can generalize to vastly more complex scenarios than those encountered in training. We introduce the concept of "complexity out-of-distribution" to characterize this challenge. We argue that scaling current deep learning approaches ("System-1") is insufficient. Instead, we propose a complementary "System-2" framework inspired by human reasoning, which frames problem-solving as a search over sequences of semantic units to represent solutions of unbounded complexity. Realizing this perspective requires developing System-2 components grounded in complexity considerations, including new benchmarks, supervision paradigms, representations, and inductive biases. While recent work shows progress towards System-2 aspects, key challenges remain in formalizing the complexity of out-of-distribution setting, developing effective System-2 learning methods, and integrating symbolic search with neural architectures. This System-1 and System-2 perspective may provide a unifying framework for transcending current AI limitations. We call for future work to translate these ideas into concrete models, advancing towards AGI.

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
