# OpenReview forum: "Position: System-2 AI is about Complexity Out of Distribution"
_ICML.cc/2025/Position_Paper_Track — Submitted to ICML 2025 Position Paper Track_

### Official Review · Reviewer_zxkb · 2025-03-12

**Significance:** 2
**Argument Clarity:** 2
**Rating:** 2
**Confidence:** 4

**Questions:**

Questions:
- In Section 5.2, the authors suggest that System-2 requires new supervised paradigms (e.g., supervision of programs or unsupervised learning), but in reality, obtaining program-level supervised data can be very difficult. How does the paper address this challenge? What are the specific strategies for unsupervised program learning?
- Is there a clear boundary between what tasks are suitable for System-1 and System-2, and if not, does this blur the applicability of the framework?
- Distilling System 2 into System 1 is another approach that bridges System 1 and System 2. Is it possible to analyze whether an approach like this could bring additions to the framework proposed in the paper?

**Discussion Potential:**

2

**Paper Summary:**

This paper advocates shifting from the "System-1" deep learning paradigm to a "System-2" reasoning framework to address AI generalization challenges, particularly "complexity out-of-distribution" (Complexity OoD). The authors propose that System-2 solves problems via searches over unbounded sequences of semantic units ("programs"), building on System-1’s foundations. They explore Complexity OoD using Kolmogorov Complexity, examine System-1/System-2 synergy, and call for tailored benchmarks, supervision, and representations. It concludes by emphasizing the integration of symbolic search with neural networks to advance toward AGI.

## update after rebuttal
The authors' rebuttal has solved my partial concerns.
However, my concerns about "Insufficient Justification for Complexity OoD Necessity" and "Lack of Explanation for Supervision Paradigms in System-2" are not well solved.
The other concern about "Insufficient Differentiation from Existing Work" is not answered.
I will maintain my rating as 2.

**Position:**

Yes

**Position In Title:**

Yes

**Related Work:**

2

**Strengths And Weaknesses:**

Strengths:
- The paper presents a novel conceptual framework. By introducing “Complexity OoD” as a distinct challenge in AI generalization, this work frames System-2 tasks as a search over unbounded sequences of semantic units. This provides a fresh perspective that bridges cognitive and machine learning.
- The use of a theoretical foundation (e.g., Kahneman Complexity) formalizes representational and computational complexity. This makes the argument more rigorous, inspiring further mathematical exploration of System-2 reasoning.
- The paper shows broad relevance and vision. It ties together diverse recent advances (e.g., Slot Attention, Chain of Thought, Tree of Thought) to System-2, positioning it as a unifying framework. This can benefit future analysis in relevant frontiers with different techniques and approaches.
- The discussion on constructing System-2 foundations (tasks, benchmarks, supervision paradigms) is concrete and forward-looking, providing actionable directions for future research.


Weaknesses:
- Lack of Concrete New Methods. The paper frames System-2 as a search over semantic units but lacks specific algorithms or architectures for implementation in neural networks, leaving the proposal abstract and impractical.
- Insufficient Justification for Complexity OoD Necessity. The need for System-2 is noted, but challenges like integrating symbolic search or managing search space complexity remain unaddressed, limiting feasibility.
- Unclear Path to Achieve System-2 and Existing Gaps. The need for System-2 is noted, but challenges like integrating symbolic search or managing search space complexity remain unaddressed, limiting feasibility.
- Weak justification for OOD generalization in Section 5.2. Section 5.2 briefly discusses supervision paradigms for System-2 but does not adequately explain why traditional approaches (e.g., supervised, unsupervised, or self-supervised learning) are insufficient for learning programs. The analogy to System-1 learning paradigms is drawn without detailed examples or theoretical grounding, making the justification feel forced and unconvincing.
- Writing style needs improvement. The writing style detracts from the paper’s clarity. For example, some paragraphs in the Introduction part lack closing punctuation, disrupting the reading flow. Section 2.1 consists of a lengthy paragraph without subheadings, making it difficult to track the progression of key ideas. Additionally, terms like “semantic units” and “programs” are used without clear definitions, obscuring key ideas.
- Insufficient differentiation from existing work. Section 2 compares complexity OoD to compositional generalization but doesn’t clearly show its unique challenges, reducing the new concept’s significance.
- Brief and Weak Discussion of Alternative Views. Section 6 introduces an alternative perspective of “learning and reasoning independence,” but the discussion is cursory and lacks rigor, making the alternative view appear as a superficial addition rather than a seriously considered perspective.

**Support:**

2

---

> ### Author Rebuttal · Authors · 2025-04-01
>
> **Lack of Concrete New Methods**
>
> As a position paper, we introduce Complexity OoD to highlight a crucial, overlooked property: current evaluations miss assessing performance on complexities beyond training data. Our value lies in defining this gap for System 2. Method invention isn't the focus of a position paper; we prioritize framing this essential problem, guiding future method development in the field for true System 2 capabilities.
>
> **Insufficient Justification for Complexity OoD Necessity**
>
> We strongly disagree that Complexity Out-of-Distribution (OoD) necessity is unjustified. The paper systematically builds this argument: Intro links multi-step System 2 problems to Complexity OoD; Section 2 differentiates it from compositional OoD; Section 4 reviews prior work relevant to Complexity OoD.  Furthermore, Section 3 addresses search space complexity by discussing System 1 heuristics to guide program search.  Regarding symbolic search challenges, we explicitly acknowledge this in Section 5 ("ground symbolic operations onto neural networks") as a key open challenge for future research, not a limitation of defining the problem itself. The paper's value lies in precisely defining this necessary problem for System 2 and outlining research directions
>
>
> **briefly discusses supervision paradigms for System-2 but does not adequately explain why traditional approaches**
>
> We believe there may be a misinterpretation of our statements regarding System 2 supervision paradigms.  We, in fact, did not claim ”traditional approaches (e.g., supervised, unsupervised, or self-supervised learning) are insufficient for learning programs.”  Our actual statement, directly drawing an analogy to System 1 learning, is precisely this: "Just like System-1, which includes supervised, unsupervised, and self-supervised learning, System-2 can also be categorized similarly”. This statement is fundamentally about highlighting the potential diversity of supervision paradigms applicable to System 2 learning, mirroring the established spectrum within System 1.
>
> **Obtaining program-level supervised data can be very difficult**
>
> We fully acknowledge the challenge of obtaining program-level supervision, and our paper explicitly addresses this point.  As stated in Section 6.2, we advocate for exploring "unsupervised programs" and propose multi-task/meta-learning on programs themselves as a promising avenue.  Learning across diverse tasks can enable the discovery of atomic units and composition rules without direct program supervision.  Furthermore, while unsupervised program learning is crucial, it's important to note that humans do learn algorithms via explicit instruction (e.g., learning the multiplication algorithm in school).  Thus, program-level supervision isn't inherently unattainable; it is possible to acquire and can be particularly effective.  While acknowledging the difficulty, our paper strategically points towards both supervised and unsupervised program learning paradigms, with a clear emphasis on the importance and potential of unsupervised approaches to overcome data acquisition challenges
>
> **Clear Boundary between what Tasks are Suitable for System-1 and System-2**
>
> The System 1/System 2 distinction isn't about inherent task properties, but model-relative, depending on the model's learned solution strategies.  Task categorization is not fixed.  Highly frequent data leads models to treat instances as "atoms"—System 1.  Less frequent data necessitates "combinations of atoms," requiring multi-step solutions—System 2.  Thus, task classification becomes dynamic and model-dependent, directly linked to data exposure and learned representations within our Complexity OoD framework
>
> **Distilling System 2 into System 1**
>
> That's an excellent question, and if you are referencing the approach in [arxiv.org/abs/2407.06023], it directly illustrates the dynamic interplay between System 1 and System 2 within our framework.  [arxiv.org/abs/2407.06023]'s method of using CoT (System 2-like) to generate training data for a direct System 1 model closely aligns with our Section 3 discussion of frequently encountered problems becoming "atoms."  This is also conceptually similar to the DreamCoder approach we discussed, where frequently co-occurring sub-programs are abstracted into new, foundational functions, effectively creating new "atoms" for more efficient problem-solving.  The key takeaway is that as a model encounters tasks frequently, its program generation modules and foundational atomic units evolve to solve those tasks with fewer steps.  In essence, increased task frequency drives a shift towards more direct, System 1-like, atomic solutions, dynamically blurring the initial System 1/System 2 task boundary and highlighting the learned and adaptive nature of this distinction within our Complexity OoD perspective.

---

### Official Review · Reviewer_9Cgt · 2025-03-13

**Significance:** 2
**Argument Clarity:** 2
**Rating:** 1
**Confidence:** 4

**Questions:**

None

**Discussion Potential:**

2

**Paper Summary:**

This paper argues that to understand and advance towards "System 2" reasoning in AI systems, we should focus on "complexity out of distribution" (COoD). System 2 is defined as problems which require multi-step reasoning to get to a solution. The paper argues that System 2 is invoked when "complexity out of distribution" is encountered. The paper defines and gives examples of COoD, including a formal definition in section 2.2 in terms of kolmogorov complexity of a new task encountered being higher than any previous task seen before. The paper outlines how System 1 and System 2 are combined to solve novel tasks, how to learn new atoms with System 1, gives a literature review of System 2 in recent research, and outlines future directions for System 2 research.

## Update after rebuttal
I maintain my score. The authors respond to each of my concerns in their rebuttal, but do not change my mind: (1) COoD as formally defined in the paper relies on conditional Kolgomorov Complexity, which, as a quantity independent of the model, does not seem to allow for variations in perceived complexity as the prior knowledge over concepts changes. Other responses do not convince me either.

**Position:**

Yes

**Position In Title:**

Yes

**Related Work:**

4

**Strengths And Weaknesses:**

I think the main presentation of System 2 as needed for tackling COoD is well supported. COoD is a catchy idea and I could see this topic being of relevance to the community and inspiring discussion. However, I don't think the evidence and argumentation supporting the idea of COoD is strong enough.

- The formal definition is a little weak: COoD is when you see a harder task than ever seen previously before? This feels a little harsh: it seems to imply that if you ever see a really hard task and learn how to solve it, suddenly nothing of lower complexity than it after that point will be COoD. A better formal definition would define a current "System 1" level and define COoD in terms of that. I'm not sure if the formal definition as presented really contributes much to the community -- it just codifies the idea of a new hard task requiring System 2.
- The discussion in Section 3 is interesting, but it mainly expounds ideas relating System 1 and System 2 that i've already seen before here and there throughout the literature. The packaging here is both a little hand-wavey, and also attached to specific examples (like programs of atoms) that make it not very insightful
- The paper concludes by suggesting that we need an ImageNet moment for System 2, and suggests that ARC might be a good benchmark to do so. I agree with this statement wholeheartedly, ARC has been the only benchmark commited to the ideas embodied by COoD! But ARC already was presented as exactly this, no need to suggest that it should happen! Moreover, ARC just got beat by OpenAI o3, which if anything is the "ImageNet" moment and validation that ARC was testing reasoning as advertised. So the position paper has a bit of "out of date" feeling to it.

- The literature review is good.

**Support:**

2

---

> ### Author Rebuttal · Authors · 2025-04-01
>
> We thank Reviewer 3 for their time and feedback on our submission. Despite their feedback, we believe a core misconception exists. We hope this rebuttal effectively clarifies our central idea and addresses their points.
>
> **Misconception of Complexity Out-of-Distribution**
>
> We must respectfully address a potential misconception to ensure a shared understanding of core contribution.  For any given instance, complexity is operationally defined as the number of steps or representational capacity necessitated for its solution by the model.  Complexity OoD thus manifests when, at test time, the model confronts instances whose complexity demonstrably exceeds the range of complexities represented in the training distribution.  Consequently, a model exhibiting overfitting to the bounded complexity inherent in the training set will predictably fail to generalize effectively to these Complexity OoD samples.
>
> **if you ever see a really hard task and learn how to solve it, nothing of lower complexity than it after that point will be COoD?**
>
> No, for two key reasons. Firstly, complexity is defined relative to the model's solution for a given instance.  A sample we perceive as "hard" might, through repeated exposure, be treated as an "atom" by the model, resulting in a shorter solution path. Conversely, a sample that appears "simpler" to us could be perceived as more complex by the model if it requires a longer, more convoluted program.  For a clearer understanding, we refer you to our Roman numeral example in response to the reviewer1.
> The second, and perhaps more salient, point is that even upon mastering a task of significant complexity, the possibility of constructing tasks of yet greater complexity is essentially limitless. An inherent characteristic of Complexity OoD is its unbounded nature.  The core significance of explicitly defining and addressing Complexity OoD lies precisely in our pursuit of verifiably ensuring that a model can robustly generalize the underlying logical principles of problem resolution to problem instances characterized by ever-increasing levels of complexity – a crucial benchmark for achieving true Artificial General Intelligence.
>
> **Concept of programs of atoms**
>
> Yes, this idea has precedents, particularly in program synthesis research.  However, in our paper, Section 3 serves a specific purpose: after introducing Complexity OoD and the need for variable-length programs to address it, we must address how such programs are constructed and how this construction can be made more efficient.  Section 3 is designed to answer this very question, bridging the conceptual gap.
>
> Furthermore, a point of potential misconception may be the interpretation of "program."  We do not necessarily mean "program" in the strict sense of algorithmically synthesized code (as in models like DreamCoder).  Instead, "program" in our context refers more broadly to the intermediate steps needed for problem resolution within System 2 reasoning.
>
> **Clarification of ARC's Role in the Context of a System 2 "ImageNet Moment"**
>
> We respectfully disagree with the interpretation that our paper positions ARC as definitively being the ImageNet moment for reasoning or that we presented it as a fully solved benchmark.
>
> Firstly, our paper does not claim ARC is the System 2 "ImageNet moment."  Instead, we explicitly suggest the need for such a benchmark and propose that ARC, with its unique design and focus on abstract reasoning, points in valuable directions for System 2 benchmark development.  We conclude that advancing System 2 research necessitates dedicated benchmarks—like ARC in spirit but potentially exceeding it in scope—specifically designed to assess program generation and Complexity OoD generalization.
>
> Secondly, while OpenAI's o3 achieving high scores on the original ARC challenge is indeed a significant advancement, it is crucial to contextualize this achievement.  It is inaccurate to portray o3's performance as definitively "beating" or "solving" ARC in the way ImageNet classification was arguably "solved" for many practical purposes in computer vision.  Several critical points underscore this distinction:
>
> - Human-Level Solution Disparity: Crucially, o3's high performance was achieved at a computational cost that is orders of magnitude greater than that of human solvers.
>
> - The recent ARC 2 release further underscores the persistent challenge of ARC-style reasoning. evaluations show even advanced models, including o3, struggle with ARC 2's increased complexity and novelty, achieving significantly lower performance despite the similar task nature to ARC 1.  This performance drop directly contradicts the idea that ARC is "beat" and confirms it remains a relevant benchmark, effectively probing AI reasoning boundaries.
>
> Thus our position is still timely. ARC advancements validate its importance, underscoring the need for more advanced benchmarks to push AI reasoning on Complexity OoD, which ARC starts addressing

---

### Official Review · Reviewer_iRTf · 2025-03-14

**Significance:** 4
**Argument Clarity:** 3
**Rating:** 3
**Confidence:** 4

**Questions:**

1) In the beginning of the paper, the authors argue that 'scaling data size would not ensure the ability to handle increased complexity'. But while this is tangentially touched upon in the rest of the paper, only in the conclusion is scaling again mentioned, saying "We argue that scaling current deep learning approaches is insufficient". Is there a part of the paper I am overlooking, or is there a place where a concrete argument is made that scaling data size woud not ensure the ability to handle increased complexity? If so, it would be good to emphasise, and elaborate on this claim, as this seems central to the paper's position.

2) It would be useful to clarify the necessity of System2 with regards to the 'Aha' moment present in DeepSeek's architecture, showing the emergence of System2 reasoning by pure RL training on a large enough model, without any need for program synthesis approaches that the authors argue for.

3) Isn't the example of multiplication (in section 2.) contrived, in the sense that existing LLMs *do* learn full algorithms for multiplying multi-digit numbers? That is, they do learn how to phrase the problem as Python code, which can then be called and executed externally. This mirrors what humans do: we can learn the algorithm, and execute it on an external system, such as a calculator, but neither humans nor LLMs can multiply multi-digit numbers with ease, suggesting a discrepancy between the asserted difference of System2 reasoning in humans, and LLMs.

Smaller questions:

3) The position stated in the first line of abstract is different than the one in the title. Do you wish to emphasise/argue a property of System2 reasoning, or emphasise/argue that System2 needs to be be introdued as a paradigm?

4) Sec 5.4 "When we built System-2...". It sounds like the authors are referring to the community building system2 approaches, and not themselves? It might be worth clarifying the intent there

**Discussion Potential:**

4

**Paper Summary:**

The paper argues that current "System-1" AI (data-driven deep learning) struggles with generalizing to scenarios requiring unbounded complexity beyond training distributions. It introduces the concept of complexity out-of-distribution (OoD), where test samples demand more intricate computational or representational steps than seen during training. To address this, the authors advocate for a complementary "System-2" paradigm, which frames problem-solving as a search over sequences of semantic units (e.g., programs) with variable complexity. Key contributions include formalizing complexity OoD via Kolmogorov complexity, proposing a program synthesis framework, and outlining research directions (tasks, benchmarks, supervision paradigms) to advance System-2 AI. The paper integrates insights from recent work on LLMs, neurosymbolic methods, and adaptive computation.

---

# Update after rebuttal

I thank the authors for their comments - I have decided to maintain my score.

Part of the reason is because of the lack of a satisfying answer to the length generalisation question. The authors stated
> Humans learn a multiplication algorithm. When correctly applied, this algorithm guarantees correct answers regardless of digit length

which is falling back to a premise I acknowledged in my original post
> ...This mirrors what humans do: we can learn the algorithm, and execute it on an external system, such as a calculator

and argued that firstly, LLMs do exactly this:

> ...they do learn how to phrase the problem as Python code, which can then be called and executed externally.

and second, that this is a problematic framing:

>  but neither humans nor LLMs can multiply multi-digit numbers with ease, suggesting a discrepancy between the asserted difference of System2 reasoning in humans, and LLMs.

The authors acknowledged LLM with backtracking on this approach has not sufficiently been explored, defeating one of the important claims of their paper

> we acknowledge that specific research on guaranteed length generalization for pure RL Transformers (like DeepSeek), especially with backtracking, hasn't be explored yet. This gap itself points to a significant, open future research direction.

This points to the fact that program synthesis approaches to this problem might not be needed, or at least that the evidence pointing to the fact that they *are* needed is weaker than posited in the paper.

**Position:**

Yes

**Position In Title:**

Yes

**Related Work:**

4

**Strengths And Weaknesses:**

### Strengths

* Clearly articulates the limitations of System-1 AI and the need for System-2 reasoning.
* Formalizes complexity OOD using Kolmogorov complexity, which I believe grounds theoretical arguments
* Provides some examples such as Roman numerals and multi-digit mulitplication to illustrate System-2 challenges.
* Really thorough discussion of recent advances (e.g., CoT, GoT, repeated sampling, RL approaches) and their alignment with System-2.
* Proposes valuable and actionable research directions: benchmarks, suggestons to redefine basic representations and metrics necessary to do it well in program search setting

### Weaknesses

See my list of questions for a list of perceived weaknesses

**Support:**

3

---

> ### Author Rebuttal · Authors · 2025-04-01
>
> Thank you to the reviewer for their careful and detailed review. We are especially pleased that they found our research directions to be both valuable and actionable.
>
> **scaling data size would not ensure the ability to handle increased complexity?**
>
> We thank the reviewer for drawing attention to this critical point. Our claim that “scaling data size اwould not ensure the ability to handle increased complexity” is indeed central to our argument.
> To elaborate, our central argument is built on the observation that even if one could vastly increase the amount of training data, there remains an inherent bound: training data can only contain examples up to a certain level of complexity. In paper, we formalize this idea through the lens of representational and computational complexity using Kolmogorov Complexity (see Sections “Representational Complexity OoD” and “Computational Complexity OoD”). These definitions show that for any finite dataset, there will always exist test cases whose minimal description or the shortest program needed to solve them is significantly more complex than what has been encountered during training. In other words, the complexity of the test examples is “unbounded” – an ever-present out-of-distribution (OoD) challenge. *Why might more complex examples be unsolvable by algorithms learned on simpler ones?* The answer is potential overfitting. The model may overfit simpler data, deriving a flawed algorithm that, while functional for simple cases, fails to generalize to more complex instances.
>
> **clarify the necessity of System2 with regards to the 'Aha' moment present in DeepSeek's architecture**
>
> Indeed, R1-like models are implicitly learning program generation, aligning with our discussion in Section 4.3 where we propose viewing LLMs as program generation modules that construct and execute programs during inference. The "aha moment" in models like R1 can be interpreted as the LLM, during inference, recognizing that its initially generated program is insufficient, thus necessitating backtracking and the synthesis of a revised program.  Effectively, pure RL methods like R1 enable the LLM to inherently understand that more complex problems demand more extensive computation and program construction.
>
> Another compelling aspect for assessing pure RL methods like R1 in the context of Complexity OoD lies in evaluating their efficiency and the program length (number of steps) they generate.  As pure RL methods like R1 are essentially learning inference-time search strategies, we can draw an analogy to standard RL evaluation. Just as the efficiency of other RL methods is measured by how fast they can find the solution, a valuable avenue for evaluating R1-like methods is to train them on examples with bounded complexity, and subsequently test their inference performance on examples exhibiting higher Complexity OoD.
>
> **LLMs do learn full algorithms for multiplying multi-digit numbers?**
>
> For multiplication, LLMs essentially have two distinct approaches: either generating and executing Python code or attempting to perform the calculation internally, without external tools. As highlighted in length generalization studies, a crucial difference emerges: humans, given sufficient attention and working memory, can accurately perform mathematical operations reliably, exhibiting robust length generalization in mathematical reasoning.  LLMs, however, do not possess this same guaranteed length generalization, particularly for complex or lengthy mathematical problems when relying solely on internal computation. This is because LLMs, instead of learning the underlying logic of multiplication – a logic inherently generalizable to numbers of any length – primarily learn to mimic the process as observed in their training data.  They are, in essence, pattern-matching procedural steps rather than grasping the abstract mathematical principles themselves.  This contrasts sharply with human mathematical understanding, which is built upon a foundational grasp of logical structure that ensures generalizability.
>
> **The position stated in the first line of abstract is different than the one in the title?**
>
> You raise a pertinent point. Indeed, the positions articulated in the title and abstract are intentionally unified. The core elements of our position are "System 2" and "Complexity Out-of-Distribution" (OoD).
>
> The position presented in the abstract is structured in two key parts.  Firstly, it asserts that the fundamental nature of the Out-of-Distribution challenge underlying "System 2" problems is precisely Complexity OoD.  Secondly, it posits that effectively addressing and overcoming these Complexity OoD scenarios necessitates complexity-aware solutions. These solutions must be equipped to handle increased complexity in both representation and computation, implying a need for models capable of adapting to varying levels of representational richness and computational depth.

---

> > ### Comment · Reviewer_iRTf · 2025-04-06
> >
> > I thank the authors for their detailed answers.
> >
> > **Scaling data size**
> >
> > It would be great if this was elaborated made more clear from the text in the paragraphs where they mention it.
> >
> > **Aha-moment**
> >
> > It sounds like the authors are agreeing that the DeepSeek Aha moment arose without any built in System2 methods. Wouldn't that mean that System1 methods might be all we need?
> >
> > **Digit multiplication**
> >
> > You stated two claims:
> > 1) "humans, given sufficient attention and working memory, can accurately perform mathematical operations reliably, exhibiting robust length generalization in mathematical reasoning."
> > and
> > 2) "LLMs, however, do not possess this same guaranteed length generalization, particularly for complex or lengthy mathematical problems when relying solely on internal computation."
> >
> > Your first claim seems to involve a high level tolerance to error, or very high outliers compared to average humans: I find it difficult to imagine a human who can multiply two 100 digit numbers easily, even with very good attention and working memory. But this is not problematic in itself - perhaps the mode of comparison is that we allow error, mistakes, and backtracking? That I would believe.
> >
> > In that case, I would urge the authors to provide some evidence of the second claim in the paper. This means both evidence showing LLMs fail to one-shot complex mathematical calculations, but also evidence showing LLMs fail at those even with some tolerance of backtracking and revision. I am personally not aware of any results of the latter form.

---

> > > ### Author Response · Authors · 2025-04-09
> > >
> > > We sincerely appreciate Reviewer iRTf's continued engagement with our work and thank them for reading our rebuttal and posing these additional insightful questions.
> > >
> > > **Aha Moment: System 1 or System 2?**
> > >
> > > The reviewer raises a very important question about the DeepSeek "Aha moment" and System 1 vs. System 2.  To clarify: we believe it is crucial to distinguish between Supervised Learning (SL) and Reinforcement Learning (RL) in this context.  While both are forms of "learning,"  it’s inaccurate to equate RL outcomes with "System 1 methods" as typically understood in the System 1/System 2 framework.  Even though DeepSeek R1 is trained via RL (a learning paradigm), its approach inherently embodies System 2 characteristics. RL, in its very essence, requires step-by-step solution construction, active exploration of different pathways, and the capacity to adapt strategies—all hallmarks of System 2 reasoning.  Therefore, the DeepSeek "Aha moment," arising within an RL framework, is more accurately interpreted as a System 2-like phenomenon (emergent through RL processes), rather than evidence that System 1 alone suffices
> > >
> > > **Human Length Generalization on Digit multiplication:**
> > >
> > > The reviewer’s question regarding 100-digit multiplication crisply highlights the core difference.  Regarding human math "reliability": it’s not about being error-proof with colossal calculations, but about the underlying algorithmic basis of human math. Humans learn a multiplication algorithm. When correctly applied, this algorithm guarantees correct answers regardless of digit length (allowing for error correction).  Humans might err in 100-digit sums due to execution mistakes—but they are, in principle, algorithmically capable and can self-correct because they possess algorithmic knowledge.
> > >
> > > **Pure-RL Methods Generalization:**
> > >
> > > We thank the reviewer for highlighting this important nuance about empirical evidence. While Transformer length generalization limitations are well-documented ([arxiv.org/abs/2207.04901],[arxiv.org/abs/2402.09371],[arxiv.org/abs/2307.03381]), we acknowledge that specific research on guaranteed length generalization for pure RL Transformers (like DeepSeek), especially with backtracking, hasn't be explored yet.
> > > This gap itself points to a significant, open future research direction.  Directly comparing length generalization in pure RL Transformers versus other models is crucial.  Investigating if pure RL models (like DeepSeek) exhibit guaranteed length generalization, and contrasting their backtracking efficacy with other architectures, is a key open avenue—valuable for differentiating model capabilities and charting future progress in addressing Complexity OoD generalization. Crucially, even if pure RL models could achieve guaranteed length generalization, the amount of backtracking required would itself become a vital metric for comparison, highlighting computational efficiency and differing problem-solving strategies

---

### Official Review · Reviewer_oR8p · 2025-03-17

**Significance:** 3
**Argument Clarity:** 2
**Rating:** 2
**Confidence:** 3

**Questions:**

What is the normative consequence of researchers using kolmogorov complexity rather than concepts from compositionality literature?

**Discussion Potential:**

2

**Paper Summary:**

This paper proposes that "System-2" AI should be expressed in terms of the kolmogorov complexity of the train vs test examples. A reasoning system should be able to handle arbitrarily complex test examples. They say that this is different from the notion of productivity because productivity is only about the length of the sequence, rather than other elements that contribute to its complexity.

## update after rebuttal

This position paper doesn’t have clear normative implications. I encourage the authors to write a main track paper using this framework to demonstrate what results are achievable with it and not with other frameworks like compositionality.

**Position:**

Yes

**Position In Title:**

Yes

**Related Work:**

2

**Strengths And Weaknesses:**

The argument for complexity as underlying test of reasoning potentially opens a path connecting information theory and compositionality research. I find the general idea appealing, and the authors are clear about the connection complex reasoning and the requirement to generalize to more complex test time examples.

This paper has a good literature review that includes more older work than most current papers do, though it is a little sparse on newer work.

What are the normative implications of switching from a compositionality perspective to a kolmogorov complexity perspective? The advantage of compositionality is that we can explicitly describe the exact concepts being composed. How could this complexity metric be incorporated, in practice, into research? What are the implications of this alternative framework? Unfortunately, the section on how this should change research is the weakest and least clear part of the paper.

And what about when models fail moving from complex to simple scenarios, e.g., “illusion illusions” (when models treat simple questions that mirror optical illusion descriptions as though the images are optical illusions rather than simple examples)? It is possible for simple examples to be out of distribution where complex examples are not. For a human, straightforward questions are simpler than describing optical illusions, but the model simply draws on its learned assumptions about how optic illusions are described. In this case, the simplest way to describe it is that complexity increases because the model has learned a simple representation of optical illusions, and then has to reason its way out, but then it may not be described as shorter in the “shortest program”.

> Adaptive computation time

It’s good to see the authors provide a literature review that covers early work on dynamic inference time. However, I’m surprised to see some of the most recent conversations missing. What about more recent advances in test time scaling, particularly the response to DeepSeek R1? Thought trace approaches have introduced new ways of scaling reasoning during inference.

> since LLMs  are still confined to left-to-right decision-making processes (without backtracking) during inference,

Recent research has indicated that backtracking is key to current LLM reasoning, from the “aha” moments that are accessible even before RL training  (https://oatllm.notion.site/oat-zero) to the finding that forcing backtracking prior to RL training allows models to learn reasoning faster (https://arxiv.org/abs/2503.01307).

When the paper brings up ARC (Abstraction and Reasoning Corpus), it is unclear how the tasks, which are designed to be complex due to their few-shot nature, align with a model in which test time complexity is the distinguishing feature of reasoning. Furthermore, humans can usually quickly select the correct output without having to explain their reasoning, suggesting that it may be described as System-1 behavior. What is the relevance of this benchmark system to the proposal.

The paper falls through when it tries to provide normative proposals for how this view of complexity should change the research landscape. What distinguishes a system-1 and system-2 supervision paradigm through the lens of complexity? What distinguishes the required tasks and benchmarks, through the lens of complexity? It is extremely difficult to measure kolmogorov complexity, even if one accepts the position of the paper that it’s the correct lens to view “System-2”. It is difficult to understand the position of this paper, without specific recommendations for how incorporating new complexity metrics should change the research landscape.

MINOR:

repeated use of “ in latex rather than ``, leading to misformatted quotes.

“some shines of system-2” what does this mean? It’s not in idiom that I’m familiar with.

> comparing the square root of 3 to that of 16.

I assume this means that the square root of three is more complex than sixteen, but you should make this explicit because it's not clear from the ordering. In this case, however, the thing that makes sqrt(3) more complex is fundamentally the length of the output or of the reasoning steps, so arguably this is a situation described by productivity. Another relevant facet of compositionality described by Hupkes et al. is localism, a way of building meaning hierarchically and thereby generalizing to tree structures of arbitrary complexity.

Missing reference: I recommend taking a look at https://www.nature.com/articles/s42256-023-00729-y to inform your discussion of complexity based train/test splits.

**Support:**

3

---

> ### Author Rebuttal · Authors · 2025-04-01
>
> We are truly grateful to Reviewer for their exceptionally thorough and careful reading of our paper.  We are particularly pleased and encouraged by their insightful recognition that our work offers a promising path toward connecting the fields of reasoning and compositionality with information theory.
>
> **implications of switching to a complexity perspective?**
>
> Intelligent OoD generalization, like humans, depends on exploiting appropriate inductive biases.
> We argue that problems categorized as "System 2" in the literature exhibit a distinct type of OoD, which we term "Complexity Out-of-Distribution".  This is defined as: if a model can solve a problem, it should also be able to solve the same problem when presented with instances requiring more computation or a richer representation.
>
> Complexity OoD, by its definition, poses a significant challenge to System-1 approaches, often characterized by limited representational and computational capacity, and System-2 solutions are necessary for them.  Furthermore, by explicitly defining Complexity OoD, we effectively bridge the gap between reasoning and learning-based approaches. Analogous to how the concept of OoD has shaped research in learning, we posit that Complexity OoD provides a valuable framework for analyzing and addressing generalization challenges specifically within the domain of reasoning.
>
> *Why not compositional perspective?* Compositionality which typically studied through the lens of  systematicity (generalization to novel combinations) and productivity (generalization to longer sequences) are often defined in terms of novel combinations in input/output spaces.
> In contrast, Complexity OoD fundamentally focuses on the representational or computational complexity inherent in solving a given problem instance. Consequences: Benchmarks prioritize escalating difficulty, not just novelty. Models must handle unbounded complexity, demanding adaptive architectures (variable compute/representation).  Evaluation measures complexity-handling efficiency, not just accuracy. This moves AI towards genuine System 2 reasoning and AGI by explicitly tackling the complexity* challenge, often missed by compositionality alone in OOD generalization
>
> **what about when models fail moving from complex to simple scenarios?**
>
> This is an excellent question that highlights a crucial nuance concerning the selection of atomic units and program construction.  a model might, due to biases in the training data, represent what we consider complex examples with seemingly "shorter programs" simply because they are encountered more frequently.
> Let's consider Roman numeral example again.  While conceptually, the quantity "ten" (10) might be considered more complex in terms of counting steps than "eight" (8), the Roman numeral system, due to greater exposure to and prevalence of "ten," encodes it as a single atom "X." In contrast, representing "eight" (8) requires combining three atoms: "VIII."
> Thus, this actually reinforces the necessity for a Complexity OoD perspective precisely because it reveals a nuanced form of out-of-distribution generalization that input/output-centric frameworks like productivity and systematicity might miss.
>
> **some of the most recent conversations missing and Relation to Pure RL**
>
> Regarding DeepSeek R1: its very recent release before submission made thorough analysis impossible.  However, we recognized its relevance, briefly mentioning its pure RL approach in (Subsection LLM and RL) to acknowledge these developments.
> Pure RL approaches like DeepSeek R1, actually strongly underscore the relevance of our Complexity Out-of-Distribution framework.
> These approaches inherently learn to allocate computational resources proportionally to the problem's complexity during inference.  The "aha moment" itself can be interpreted as precisely the point at which the model recognizes the need to invest greater computational effort to solve a particular instance.
> Regarding the critique of our statement "since LLMs are still confined to left-to-right decision-making processes":  This statement was specifically made within the context of discussing (CoT) methods. Our intention was to highlight the limitation of naive CoT in certain reasoning scenarios and to contextualize the motivation for more sophisticated methods like ToT.
>
> **ARC**
>
> what makes ARC particularly compelling and directly relevant to our proposal is the private nature of its test set, in addition to its need for program synthesis .  This privacy creates a situation akin to developers having no prior knowledge about the complexity of the test examples. so developers are effectively forced to design models capable of handling Complexity Out-of-Distribution.

---

> > ### Comment · Reviewer_oR8p · 2025-04-01
> >
> > Thanks for your response. I'm still not clear on what the implications of moving from a compositional perspective would be, in terms of practical predictions. In general, I don't see the normative implications of this switch. I think that the position would be better served by a non-position research paper that could illustrate some kind of improvement in theoretical modelling or empirical prediction about model behavior. I will maintain my score.

---

### Decision · Program_Chairs · 2025-04-27

**Decision:**

Reject

**Comment:**

This paper proposes system-2 AI should be expressed via Kolmogorov complexity. This paper definitely addresses a long-term discussion on how to represent system-2 and what is the relationship between it and system-1. The paper provided new theoretical view to modeling system-2 via complexity OOD. It also provided a good discussion of recent advances and their connections to System-2. But the reviewers are not fully convinced by the new view, and arguing whether system-2 is really needed if we already see something similar from System-1. The reviewers have acknowledged reading the rebuttals. Two of them posted rebuttal comments, but are not fully convinced by the responses. They are calling for more substantial evidence for the new theory.